# Morphometrics as a Complementary Tool in the Differentiation of Two Cosmopolitan Flea Species: *Ctenocephalides felis* and *Ctenocephalides canis*

**DOI:** 10.3390/insects13080707

**Published:** 2022-08-05

**Authors:** Angela María García-Sánchez, Antonio Zurita, Cristina Cutillas

**Affiliations:** Department of Microbiology and Parasitology, Faculty of Pharmacy, University of Seville, Profesor García González 2, 41012 Sevilla, Spain

**Keywords:** fleas, Siphonaptera, *Ctenocephalides*, morphometrics, PCA, geometric morphometrics

## Abstract

**Simple Summary:**

Members of Siphonaptera are commonly known as fleas. With more than 2500 species described worldwide, they constitute one of the most important parasites in our environment. The cat flea, *Ctenocephalides felis*, and the dog flea, *Ctenocephalides canis*, can also affect humans and represent a potential danger for the transmission of pathogens. Despite being two of the most frequently studied species, the classification and taxonomic diversity of these fleas is controversial. Variations in their morphological characteristics frequently hinder their correct identification and give rise to several uncertainties. To provide further information on the identification of these flea species, a geometric morphometrics analysis was conducted. This technique assisted in differentiating between specimens of both species, demonstrating that it can provide useful complementary data and new insights for the classification of flea species, especially when molecular biology techniques are not affordable or available.

**Abstract:**

Fleas (Siphonaptera) are one of the most important ectoparasites that represent a potential danger for the transmission of pathogens in our environment. The cat flea, *Ctenocephalides felis* (Bouché, 1835), and the dog flea, *Ctenocephalides canis* (Curtis, 1826) are among the most prevalent and most frequently studied species throughout the world. However, the variations observed in their morphological characteristics complicate their correct identification, especially when there is a lack of access to the equipment and funds required to carry out molecular biology techniques. With the objective to provide an additional tool to help in the differentiation of *Ctenocephalides* species, a principal component analysis was carried out for the first time in the present work on populations of *C. felis* and *C. canis* from countries in three continents, namely Spain (Europe), South Africa (Africa) and Iran (Asia). The factor maps assisted in the differentiation of both species and the detection of differences in overall size, although morphological ambiguity prevented the delimitation in populations of the same species. Thus, morphometrics represents a complementary tool to other traditional and modern techniques, with great potential to assist in the differentiation of fleas, particularly species that have historically been difficult to identify.

## 1. Introduction

With more than 2500 species described worldwide, fleas (Siphonaptera) are one of the most important ectoparasites in the world, associated with a wide variety of hosts, and environmental and biological patterns [1]. In addition to being able to provoke itching bites and allergic skin diseases, fleas can also act as a vector for other parasites and microorganisms such as viruses and bacteria [2,3]. Thus, the presence of fleas in our environment represents a potential danger for the transmission of pathogens [1] and, for this reason, controlling them is a costly process [4]. In order to develop effective control and prevention measures, it is essential to deepen the understanding of the taxonomy and systematics of fleas associated with humans and companion animals [5,6,7].

The cat flea, *Ctenocephalides felis* (Bouché, 1835), and the dog flea, *Ctenocephalides canis* (Curtis, 1826), represent the majority of fleas infesting not only dogs and cats, but also other warm-blooded animals and even humans [2,8,9,10]. Specifically, *Ctenocephalides felis,* is the most prevalent species throughout the world with high infestation rates. This cosmopolitan distribution is due to their high adaptability to a wide variety of environmental conditions [11].

Due to their global importance and their ability to vector pathogens such as *Rickettsia felis* and *Bartonella* spp. [3,12], *C. felis* and *C. canis* are well-studied fleas through morphological and molecular techniques, which usually allow for the differentiation of both species [13,14,15]. However, variations in morphological characteristics were observed among these fleas, hindering their correct identification, and giving rise to several uncertainties about its taxonomic diversity [16]. The cat flea species historically includes four geographically defined subspecies, but their remarkable morphological ambiguity and the assumption of interbreeding between subspecies make differentiation even more complex if not impossible [5,9,17,18]. In addition, the scarcity of available genetic data for taxa in the genus *Ctenocephalides* causes their genetic identity to remain elusive [9,19,20].

On the other hand, traditional methods for diagnostics in parasitology are subject to interpretation bias and resource-poor clinical settings do not always employ the required tools and skilled technicians for data analyses. Nowadays, although molecular biology includes widespread techniques, such techniques are not always available within all laboratories. These limitations have fostered the appearance of new and more accessible techniques for data treatment, such as geometric morphometric analysis [21,22].

Geometric morphometric analysis is a novel approach of parasitological diagnosis, and it is applied to *Fasciola* spp. [23], nematodes [24] and arthropods [25,26], including fleas from the genera *Pulex* [27], *Ctenophthalmus* [28] and *Stenoponia* [29].

With the objective of offering an additional tool to help in the differentiation of *Ctenocephalides* species, a principal component analysis was carried out for the first time in the present work on populations of *C. felis* and *C. canis* in countries across three continents, namely Spain (Europe), South Africa (Africa) and Iran (Asia). On the one hand, we tried to explore the capacity of this approach to discriminate between both flea species and on the other hand between populations of the same species, as well as the potential contribution of other traditional and modern techniques.

## 2. Materials and Methods

### 2.1. Collection of Samples

A total of 246 fleas (107 males and 139 females) were collected from dogs (*Canis lupus familiaris*) from different regions of Europe, Africa and Asia, specifically Spain, South Africa and Iran, respectively, which were distributed as shown in Table 1.

Each infested dog was exhaustively examined for fleas and combed for 5 min over the whole body with a fine-toothed comb, specifically the head, neck, body, sides, tail, and ventral regions of each animal. Fleas were collected manually, transferred to Eppendorf tubes containing 96% ethanol and stored at room temperature until processing. The transportation and conservation of samples did not require any additional conditions.

### 2.2. Morphological Identification and Metric Data Processing

For the morphological analysis, all whole specimens were examined and photographed under an optical microscope to carry out a first specific classification. Subsequently, all the specimens were cleared with 10% KOH, prepared and mounted on glass slides using conventional procedures with EUKITT mounting medium (O. Kindler GmbH & Co., Freiburg, Germany) [30]. Once mounted, they were examined again for a deeper morphological analysis using a CX21 microscope (Olympus, Tokyo, Japan). Diagnostic morphological characteristics of all the samples were studied by comparison with figures, keys and descriptions reported previously [1,9,31,32,33]. After morphological identification, the cleared and mounted specimens were measured using a Zeiss microscope 47 30 11 9901 (Zeiss, Oberkochen, Germany) according to 10 different parameters for males (Table 2) and 14 different parameters for females (Table 3). These parameters were selected and measured in the present work in accordance with the representative characteristics of *Ctenocephalides* mentioned in the literature [1,9,31,32,33]. Figure 1 shows a diagram representing the biometric characteristics analyzed.

Descriptive univariate statistics based on arithmetic mean, standard deviation, range and coefficient of variation for all parameters were determined for male and female populations. The data were subjected to one-way ANOVA (analysis of variance) for statistical analysis of the parameters. The results were statistically significant when *p* < 0.05. Statistical analysis was performed using Microsoft Excel 2016 (v16.0). In addition, biometric characteristics of fleas were compared, and the most significant parameters were assayed for a morphometrics study.

Morphological variation is quantified using geometric morphometrics [21], a technique offering an estimate of size by which different axes of growth are integrated into a single variable (the “centroid size”) [34]. The estimate of size is contained in a single variable reflecting variation in many directions, correlated with the number of landmarks under study, and shape is defined as their relative positions after correction for size, position and orientation. With these informative data, and the corresponding software freely available to conduct complex analyses, significant biological and epidemiological features can be quantified more accurately [35].

Multivariate analyses were applied to assess phenotypic variations among the samples, using size-free canonical discriminant analysis on the covariance of log-transformed measurements. These analyses were applied to exclude the effect of within-group ontogenetic variations by reducing the effect of each characteristic on the first pooled within-group principal component (a multivariate size estimator) [36]. The Principal Component Analysis (PCA) was used to summarize most of the variations in a multivariate dataset with few dimensions [37]. Morphometric data were explored using multivariate analysis in four parameters (TL, TW, HW and AW) in males (Table 2) and four parameters (HL, BULGAL, APEHILL, and DEG) in females (Table 3) with BAC v.2 software [38,39].

Molecular data were analyzed previously by Marrugal et al. [18], which confirmed the morphological identification of the samples.

## 3. Results

A total of 246 fleas were collected and classified as follows: 175 from Spain as *C. felis* (76 males and 99 females), 36 from South Africa as *C. felis* (15 males and 21 females) and 10 from Iran as *C. felis* males plus 25 *C. canis* (6 males and 19 females) (Table 1).

To carry out the classification of the *Ctenocephalides* samples, we considered traditionally used descriptions to discern between these species and, additionally, detected remarkable morphological features based on the measurements performed. Statistical tests showed several significant measurements for subsequent morphometric analyses. Therefore, the following parameters were used: total length (TL), total width (TW), total width of the head (HW) and apex width (AW) in males (Table 2) and total length of the head (HL), total length of the bulga (BULGAL), total length of the apex of the hilla (APEHILL) and difference between the first and second spines of the genal ctenidium (DEG) in females (Table 3). The study of the influence of the size was carried out by performing PCA in *C. felis* and *C. canis*, consisting of the regression of each character separately on the within-group first principal component (PCI). The resulting factor maps for male and female populations are represented in Figure 2 and Figure 3, respectively.

Male variables significantly correlated with PCI, contributing 63% to the overall variation. The male factor maps did not show any remarkable global size differences in the *C. felis* populations, but a slightly larger size was detected in *C. canis* males (Figure 2). In addition, male populations presented an extensive overlapping area except for *C. canis*. This flea only overlapped partially with *C. felis* from Spain and South Africa, and it appeared completely independent from the *C. felis* population from Iran.

On the other hand, female variables significantly correlated with PCI, contributing 55% to the overall variation. The resulting factor maps (Figure 3) clearly illustrate global size differences in the populations analyzed, including a bigger size in *C. canis,* more remarkable than in males. Although the female populations showed an overlapping area, two delimited zones can be distinguished, whereby one zone is constituted by *C. felis* from Spain while the other zone consists of *C. canis* from Iran. Moreover, the *C. felis* from South Africa presents an intermediate size between *C. felis* from Spain and *C. canis* from Iran, with its own morphometric pattern. These results reveal that intermediate forms between *C. felis* and *C. canis* exist in South Africa.

## 4. Discussion

Despite their names, cat and dog fleas are not specific to either animal, as both species can be found on either a dog or a cat. In fact, Dobler and Pfeffer [40] showed that the most prevalent flea species found globally in domestic dogs is *C. felis*, with prevalence rates ranging from 5% to 100%. This is why the sampling process could be focused on dogs only.

On the other hand, the reason for the low number of *C. canis* specimens reported in Table 1 is that this species is present globally but in lower rates than the cat flea [40]. *C. felis* is the most prevalent flea species detected on dogs and cats in Europe and other regions. In Spain, *C. felis* is the most frequently detected and widely distributed throughout the country [41]. In addition, *C. canis* is considered very rare in South Africa [1], leading to a lower detection of these specimens.

The taxonomy of *Ctenocephalides* fleas remains unresolved due to complex factors such as the host range, vicariance and climatic events [1]. Finding representative parameters that assist in the differentiation of these fleas represents an elusive task, which becomes even more complex when taking subspecies into account. For instance, van der Mescht et al. [1] carried out one of the few principal component analyses applied to *C. felis*, based on the variation of the head shape between *C. f. strongylus* and *C. f. felis*. The large overlap observed in the factor map indicated that this characteristic is not useful for phylogenetic inferences. Moreover, these authors found that neither sex differed in body size between subspecies or genetic clusters.

The *C. felis* morphological ambiguity brought to light by other authors [9,19,20] could explain the overlap between these populations in the male and female factor maps in the present work (Figure 1 and Figure 2) and the lack of significant differences among *C. felis* populations, which prevent their differentiation.

In addition, Lawrence et al. [16] reported that *C. felis* was most phylogenetically diverse in Africa, with genetic assemblages that do not belong strictly to any subspecies designations. This fact is in accordance with the intermediate size presented by *C. felis* from South Africa in females (Figure 3), whose factor map overlaps with both *C. felis* from Spain and *C. canis* from Iran.

Moreover, *C. canis* presented a larger size in both male and female populations, considering the selected measurements, and showed a morphological identity that allowed its differentiation from *C. felis* in both male and female populations. 

Therefore, it is necessary to emphasize that the differentiation between *C. felis* from distinct populations seems impossible at a morphological level exclusively, and between *C. felis* and *C. canis* there may arise confusion too, even when relying on apparently trustworthy features. Complicating this further, these features also vary between genders, as evidenced by the fact that the measurements in the present study were completely different between males and females, with TL, TW, HW and AW in males versus HL, BULGAL, APEHILL and DEG in females. This is in accordance with Linardi and Santos [34], who highlighted that although the head curvature is highly different between males and females of *C. felis*, this feature may be unclear for separating males of the two species. This led to an incorrect diagnosis in some studies, in which males of *C. felis* were identified as *C. canis*. In fact, head length (HL) was not useful to discriminate between *Ctenocephalides* sp. males in the present work, as opposed to females.

It is also remarkable that all combinations of measurements for female *Ctenocephalides* in this study that included the length of the fleas always led to factor maps with wide overlapping areas between them (data not shown) meaning that the total length of the fleas (TL) does not contribute to species differentiation in females, while in males TL appeared as a significant feature.

On the other hand, the inclusion of the apex width in the PCA carried out in the present work is in accordance with the importance this parameter has shown previously to define the morphological identity in males [14,18], just as the degree of elongation of the apical part (hilla) in females [18,31]. In case of not being able to obtain apex related measurements for geometric morphometric analyses, an alternative consists of using DEG instead in males, since it proved to be a useful parameter with similar results (data not shown).

Although the separation of *C. felis* specimens from different regions was not accomplished in the present work, geometric morphometrics emerged as a complementary technique which can differentiate between *C. canis* and *C. felis*, with factor maps that highlight the differences between both fleas, relying on statistically significant morphological features. 

Recently, morphometrics has proven useful to discern between flea populations, such as *Ctenophthalmus baeticus boisseauorum* and *Ctenophthalmus apertus allani* [29] as well as *Stenoponia tripectinata tripectinata* specimens from Canary Islands and the Iberian Peninsula [30]. This technique represents an interesting approach to apply to other congeneric flea species and doubtful cases, in which the morphological features are not valid criteria as diagnostic characteristics. This is the case of the *Archaeopsylla* [42] and *Nosopsyllus* [43] species, in which the taxonomic similarity between species complicates their identification based exclusively on morphological characteristics.

The results obtained by morphometrics are supported by software analyses, hence they are more accurate than traditional techniques and, in addition, more affordable in low-resource settings [22].

Despite the limited number of *C. canis* specimens analyzed, differentiation between species and was achieved and conclusions were reached. However, it would be desirable to include a greater number of this elusive flea in future studies.

Considering that DNA sequencing techniques are costly and special equipment is required, morphometrics arise as an affordable additional criterion for systematic studies on fleas. This method also shows potential for application in other flea species that are not easy to differentiate between with traditional methods, offering new possibilities in this field.

## 5. Conclusions

The principal component analysis of males and females *C. felis* and *C. canis* revealed factor maps that allowed for the differentiation of both species, although overlapping between populations was present probably due to the morphological ambiguity of *C. felis*. Differences in overall size were also detected, with *C. canis* presenting a larger size in all cases.

Hence, the results obtained reveal that morphometrics can provide useful complementary data to delineate *Ctenocephalides* species, especially when there is no access to molecular biology techniques.

Accordingly, morphometrics represents an alternative to other traditional and modern techniques, showing an extrapolation capacity, great potential to help in the differentiation of fleas and applicability to species that have historically been difficult to identify.

## Figures and Tables

**Figure 1 insects-13-00707-f001:**
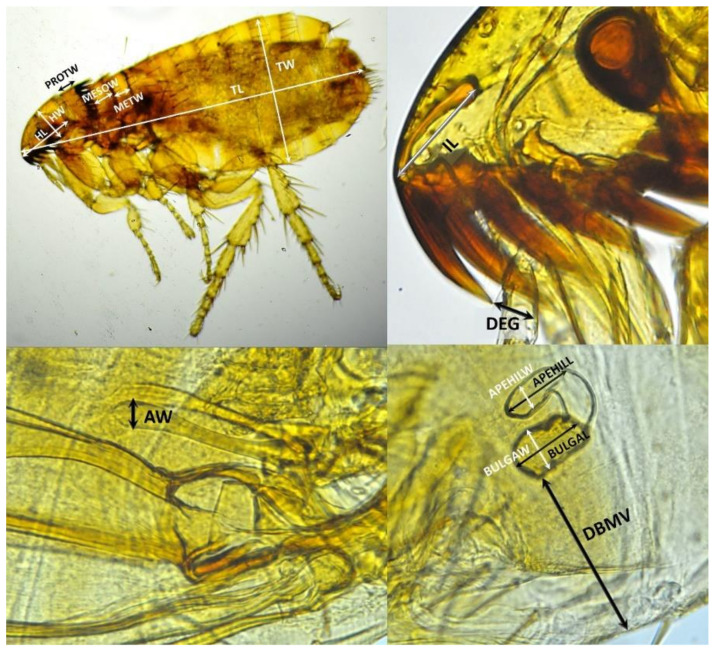
Diagram of the biometric characteristics analyzed. In all specimens: TL: total length, TW: total width, HL: total length of the head, HW: total width of the head, PROTW: total width of the prothorax, MESOW: total width of the mesothorax, METW: total width of the metathorax, DEG: difference in length between first and second spines of the genal ctenidium, IL: incrassation length from the head. In males: AW: Apex width. In females: BULGAL: total length of the bulga, BULGAW: total width of the bulga, APEHILL: total length of the apex of the hilla, APEHILW: total width of the apex of the hilla, DBMV = distance from bulga to ventral margin of the body.

**Figure 2 insects-13-00707-f002:**
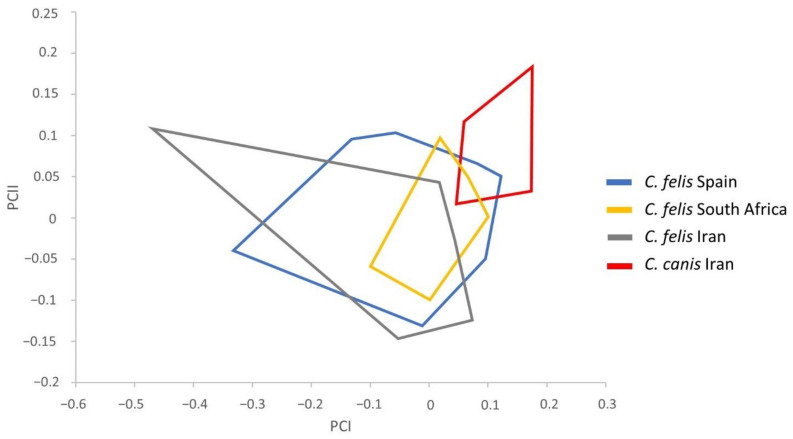
Factor map corresponding to *Ctenocephalides* spp. male adults from Spain, South Africa, and Iran. Samples are projected onto the first and second principal components: PCI (63%) and PCII (25%). Each group is represented by its perimeter.

**Figure 3 insects-13-00707-f003:**
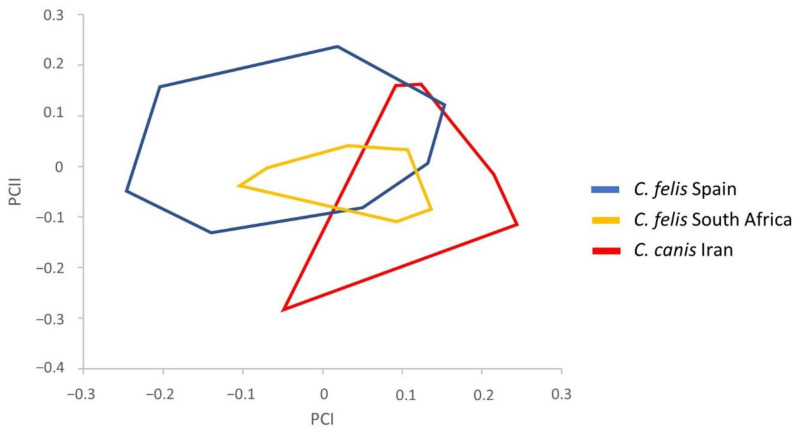
Factor map corresponding to *Ctenocephalides* spp. female adults from Spain, South Africa, and Iran. Samples are projected onto the first and second principal components: PCI (55%) and PCII (28%). Each group is represented by its perimeter.

**Table 1 insects-13-00707-t001:** Distribution of fleas collected from dogs from different geographical origins.

Geographical Origin	*Ctenocephalides felis* (Number of Fleas)		*Ctenocephalides canis* (Number of Fleas)	
	Male	Female	Male	Female
Sanlúcar de Barrameda (Cádiz, Spain)	20	25	-	-
Santanyi (Mallorca, Spain)	20	20	-	-
Fuentes de Andalucía (Seville, Spain)	-	10	-	-
Mairena del Aljarafe (Seville, Spain)	8	4	-	-
Dos Hermanas (Seville, Spain)	12	15	-	-
La Luisina (Seville, Spain)	-	10	-	-
Seville (Seville, Spain)	16	15	-	-
Polokwane (Limpopo province, South Africa)	15	21	-	-
Nashtarood (Mazandaran province, Iran)	10	-	6	19
Total	101	120	6	19

**Table 2 insects-13-00707-t002:** Biometrical data of males of *Ctenocephalides felis* and *C. canis* isolated from *Canis lupus familiaris* from Spain, South Africa and Iran.

	*C. felis* (Spain)					*C. felis* (South Africa)					*C. felis* (Iran)					*C. canis* (Iran)				
	Max	Min	Б	σ	VC	Max	Min	Б	σ	VC	Max	Min	Б	σ	VC	Max	Min	Б	σ	VC
TL (mm) †	2.2	1.5	1.8	0.2	11	2.1	1.5	1.9	0.2	11	1.9	1.4	1.7	0.2	12	2.5	1.7	2.0	0.3	15
TW (mm) †	0.9	0.6	0.8	0.1	13	0.9	0.6	0.8	0.1	13	0.9	0.6	0.7	0.1	14	1.1	0.8	0.9	0.1	11
HL (μm)	440	316	378	25	7	404	334	367	22	6	398	263	357	39	11	474	322	379	52	14
HW (μm) †	270	193	232	16	7	246	188	225	19	9	258	210	239	15	6	305	229	269	29	11
PROTW (μm) †	135	70	98	14	15	111	76	88	9	10	111	70	94	15	16	123	88	106	13	12
MESOW (μm)	135	82	111	13	12	123	76	107	16	15	135	100	109	10	9	164	70	116	30	26
METW (μm) †	158	88	121	13	11	147	82	122	17	14	147	117	133	10	8	217	141	162	31	19
DEG (μm) †	63	16	44	7	16	56	38	47	5	11	49	24	37	8	22	52	43	47	4	9
IL (μm) †	99	47	65	9	14	59	42	52	6	12	68	49	55	6	11	63	42	55	8	15
AW (μm) †	94	35	75	11	15	92	61	78	8	10	94	23	64	21	33	106	80	90	11	12

TL: total length, TW: total width, HL: total length of the head, HW: total width of the head, PROTW: total width of the prothorax, MESOW: total width of the mesothorax, METW: total width of the metathorax, DEG: difference in length between first and second spines of the genal ctenidium, IL: incrassation length from the head, AW: Apex width, Max: maximum, Min: minimum, Б: arithmetic mean, σ: standard deviation, VC: coefficient of variation (percentage converted), †: Significant differences between groups (*p* < 0.005).

**Table 3 insects-13-00707-t003:** Biometrical data of females of *Ctenocephalides felis* and *C. canis* isolated from *Canis lupus familiaris* from Spain, South Africa and Iran.

	*C. felis* (Spain)					*C. felis* (South Africa)					*C. canis* (Iran)					
s	Max	Min	Б	σ	VC	Max	Min	Б	σ	VC	Max	Min	Б	σ	VC	
TL (mm)	3.0	1.7	2.4	0.3	13	2.7	2.0	2.5	0.1	4	3.4	1.8	2.4	0.4	17	
TW (mm)	1.4	0.8	1.1	0.1	9	1.2	0.9	1.1	0.1	10	1.3	0.9	1.1	0.1	9	
HL (μm) †	486	369	430	24	6	440	375	410	21	5	422	328	384	26	7	
HW (μm)	310	229	270	17	6	299	240	273	16	6	316	240	275	23	8	
PROTW (μm) †	188	88	122	19	15	147	100	117	10	8	135	70	95	16	17	
MESOW (μm)	240	105	140	20	14	164	117	137	13	9	152	105	130	14	11	
METW (μm) †	170	117	145	13	9	164	111	144	14	10	217	117	173	23	13	
DEG (μm) †	59	28	45	6	14	54	35	46	6	13	82	35	56	11	20	
IL (μm) †	118	59	88	12	13	92	49	72	11	15	94	38	72	14	19	
BULGAL (μm) †	82	47	65	7	11	78	52	66	7	11	85	52	70	10	14	
BULGAW (μm)	68	40	51	4	8	56	42	49	7	14	59	47	51	3	6	
APEHILL (μm) †	78	31	51	10	20	68	40	54	8	14	82	40	67	11	17	
APEHILW (μm) †	42	19	29	4	14	40	14	27	6	21	49	24	35	6	18	
DBMV (μm)	410	29	249	69	28	287	147	223	42	19	340	176	247	45	18	

TL: total length, TW: total width, HL: total length of the head, HW: total width of the head, PROTW: total width of the prothorax, MESOW: total width of the mesothorax, METW: total width of the metathorax, DEG: difference in length between first and second spines of the genal ctenidium, IL: incrassation length from the head, BULGAL: total length of the bulga, BULGAW: total width of the bulga, APEHILL: total length of the apex of the hilla, APEHILW: total width of the apex of the hilla, DBMV = distance from bulga to ventral margin of the body, Max: maximum, Min: minimum, Б: arithmetic mean, σ: standard deviation, VC: coefficient of variation (percentage converted), †: Significant differences between groups (*p* < 0.005).

## Data Availability

The data presented in this study are available in the article.

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
