# Peer review of "Morphometrics as a Complementary Tool in the Differentiation of Two Cosmopolitan Flea Species: Ctenocephalides felis and Ctenocephalides canis"

_insects, 2022, doi:10.3390/insects13080707_

Round 1

Reviewer 1 Report

The MS provides useful complementary data and new insights to the classification of flea species, especially in subspecies. Here I give two suggestions as follows. Firstly, in line 102-103, as for “according to 10 different parameters for males (Table 2) and 14 different parameters for females (Tables 3)” is concerned, some references should be cited. If some measure data comes from your work, you should also point out. Secondly, if the molecular data are added (unpulsory), the MS would obtain more supports to show diversity especially in the level of Ctenocephalides felis subspecies.

Author Response

The MS provides useful complementary data and new insights to the classification of flea species, especially in subspecies. Here I give two suggestions as follows. Firstly, in line 102-103, as for “according to 10 different parameters for males (Table 2) and 14 different parameters for females (Tables 3)” is concerned, some references should be cited. If some measure data comes from your work, you should also point out.

Lines 101-103 include references of the considered previous flea descriptions [1, 9, 31-33] and all measure data analyzed have been obtained in the present work. Accordingly, a new sentence has been included in lines 106-109.

Secondly, if the molecular data are added (unpulsory), the MS would obtain more supports to show diversity especially in the level of Ctenocephalides felis subspecies.

Molecular data were analyzed previously by Marrugal et al. [18], who confirmed the morphological identification of the flea samples. A new sentence has been included in Material and methods section (lines 155-156).

Reviewer 2 Report

There is a need for the following clarifies and improvements in this manuscript:

1.       If this study planned to offer a tool to help differentiate Ctenocephalides species, specifically, the cat flea, Ctenocephalides felis, and the dog flea, Ctenocephalides canis. It is unclear why specimens were collected only from dogs (Canis lupus familiaris).

2.       C. canis number of individuals is not comparable with the number obtained for C. felis. Please clarify the reason for the low number of C. canis specimens reported in Table 1?

3.       A diagram (over the flea body) representing the biometric characters analyzed would be recommended.

4.       Considering the difference between samples size obtained (table 1), please clarify the sturdiness that has the analysis performed under this affirmation "In addition, biometric characters of fleas were compared between different regions, and the most significant parameters were assayed for a morphometrics study".

5.       A description of preanalytical aspects such as collection procedure, transportation, and conservation of the biological sample is suggested.

6.       At the end of the discussion, a short paragraph would be desirable where the authors mention the limitations of this study, its impact, contributions, and research perspectives.

7.       In the conclusions section, the following affirmation is not a conclusion from this study “DNA sequencing techniques are costly and special equipment is required”.. please rephrase this section to be more specific about the findings related to this study.

Author Response

There is a need for the following clarifies and improvements in this manuscript:

  1. If this study planned to offer a tool to help differentiate Ctenocephalides species, specifically, the cat flea, Ctenocephalides felis, and the dog flea, Ctenocephalides canis. It is unclear why specimens were collected only from dogs (Canis lupus familiaris).

Despite their names, cat and dog fleas are not specific to either animal, as both species can be found on either a dog or a cat. In fact, Dobler and Pfeffer (2011) showed that the most prevalent flea species found globally in domestic dogs is the cat flea (C. felis), with prevalence rates range from 5% to 100%. Considering the objectives of the study, the sampling could be focused on dogs only. This explanation has been included in lines 208.

  1. C. canisnumber of individuals is not comparable with the number obtained for C. felis. Please clarify the reason for the low number of C. canis specimens reported in Table 1?

The dog flea (C. canis) is present globally, but in lower rates than the cat flea (Dobler and Pfeffer, 2011). C. felis is the most prevalent flea species detected on dogs and cats in Europe and other world regions. In Spain, one of the sample points of the present work, C. felis is the most frequently detected and widely distributed throughout the country (Gálvez et al., 2017). On the other hand, C. canis is considered very rare in South Africa (L. van der Mescht et al., 2021), defaulting the detection of these specimens. This explanation has been included in lines 208-213.

  1. 3.   A diagram (over the flea body) representing the biometric characters analyzed would be recommended.

A new diagram showing the characters analyzed has been included. See Figure 1. 

  1. Considering the difference between samples size obtained (table 1), please clarify the sturdiness that has the analysis performed under this affirmation "In addition, biometric characters of fleas were compared between different regions, and the most significant parameters were assayed for a morphometrics study".

Sentence has been modified (lines 124-126). 

  1. A description of preanalytical aspects such as collection procedure, transportation, and conservation of the biological sample is suggested.

 Additional information related to sampling process has been included (lines 87-93).

  1. At the end of the discussion, a short paragraph would be desirable where the authors mention the limitations of this study, its impact, contributions, and research perspectives.

A new paragraph has been added in the discussion section (lines 275-282).

  1. In the conclusions section, the following affirmation is not a conclusion from this study “DNA sequencing techniques are costly and special equipment is required”. please rephrase this section to be more specific about the findings related to this study.

Conclusion section has been modified accordingly.

Round 2

Reviewer 2 Report

The authors have adequately addressed the suggestions, well-argued their answers, and incorporated the pertinent improvements in the manuscript.